# The Expected Values for the Gutman Index and Schultz Index in the Random Regular Polygonal Chains

**DOI:** 10.3390/molecules27206838

**Published:** 2022-10-12

**Authors:** Xinmei Liu, Qian Zhan

**Affiliations:** School of Mathematics and Big Data, Anhui University of Science & Technology, Huainan 232000, China

**Keywords:** Gutman index, Schultz index, random regular polygonal chain, expected value

## Abstract

Two famous topological indices, the Gutman index and Schultz index, are studied in this article. We mainly calculate the exact analytical formulae for the expected values of the Gutman index and Schultz index of a random regular polygonal chain with *n* regular polygons. Moreover, we determine the average values and the extremal values of the indices in regard to the set of all these regular polygonal chains.

## 1. Introduction

Thus far, chemistry has been extensively studied and applied in graph theory [1,2,3,4]. It is often the case that chemical compounds can be interpreted by chemical graphs, in which vertices represent the atoms and edges stand for the covalent bonds between atoms. The random regular polygonal chains studied here are obtained by randomly connecting *n* regular polygons through vertex-to-vertex in turn; these connection vertices are termed cut vertex. Meanwhile, the random regular polygonal chains with *n* regular polygons are spiro compounds, which are a significant class of cycloalkanes in organic chemistry; refer to [5,6]. Spiro compounds have a rigid and stable structure, their chiral ligands have a large specific optical rotation, and they have important applications in asymmetric catalysis, luminescent materials, pesticides, polymer binders and so on.

In theoretical chemistry, topological indices are commonly used to encode molecules, as well as to design compounds with physicochemical properties or biological activity [7]. It is a popular issue to study the topological indices of the chemical chains and their significance.

Throughout this paper, all the graphs that we take into account are finite, undirected and simple; a graph is simple if it has no loops and no two of its links join the same pair of vertices. More detailed notations and terminologies can be found in [8]. Let G=(V(G),E(G)) be a disordered graph; V(G) represents the vertex set and E(G) stands for the edge set. The degree $d_{G}\left(u\right)$, or d(u) for short, is the number of edges connected to the node *u*(u∈V(G)). The length of the shortest path between vertex *u* and vertex *v*(u,v∈V(G)) is defined as the distance $d_{G}\left(u,v\right)$ (short for d(u,v)). The sum of the distances between any two vertices in V(G) is $d\left(u|G\right)=\sum_{v\in V(G)}d\left(u,v\right)$.

The Wiener index [9,10,11,12,13,14] is the best-known of all topological indices, which is defined as $W\left(G\right)=\sum_{\left\{u,v\}\subseteq V(G\right)}d_{G}\left(u,v\right)$. By weighting the Wiener index, the Gutman index is proposed as
(1)$$Gut\left(G\right)=\frac{1}{2}\sum_{u\in V_{G}}\sum_{v\in V_{G}}\left(d_{G}\left(u\right)d_{G}\left(v\right)\right)d_{G}\left(u,v\right)=\sum_{\left\{u,v\}\subseteq V(G\right)}\left(d_{G}\left(u\right)d_{G}\left(v\right)\right)d_{G}\left(u,v\right),$$
which is a kind of vertex-valency-weighted sum of the distance between all pairs of vertices in a graph [15,16,17,18,19]. The index is closely related to the Wiener index and has the same structured features as the Wiener index in the condition of acyclic structures. Therefore, the research for the possible chemical applications of the Gutman index and the theoretical investigations should lay particular emphasis on the condition of polycyclic molecules.

The Schultz index [20] is proposed as
(2)$$S\left(G\right)=\frac{1}{2}\sum_{u\in V_{G}}\sum_{v\in V_{G}}\left(d_{G}\left(u\right)+d_{G}\left(v\right)\right)d_{G}\left(u,v\right)=\sum_{\left\{u,v\}\subseteq V(G\right)}\left(d_{G}\left(u\right)+d_{G}\left(v\right)\right)d_{G}\left(u,v\right).$$

The Schultz index has been shown to be a useful molecular descriptor in the design of molecules with desired properties [21]; thus, further studies on the mathematical and computational properties of the Schultz index are desirable, including more articles on developing such topology indices of the mathematical properties and applications [22].

Many scholars have studied the indices of some random polyphenylene chains before [23,24,25,26,27,28]. Recently, Liu and Zeng et al. [29] obtained some indices in the random spiro chains, including the Gutman index, Schultz index, multiplicative-degree Kirchhoff index and additive-degree Kirchhoff index, and determined the expected values of these indices in the random spiro chain, and the extremal values among all spiro chain with *n* hexagons. Motivated by [29,30,31], we explore the property of the Gutman index and Schultz index of polygonal regular chains, and determine the expected values of the indices, denoted by $E(Gut\left(SC_{n}\right))$ and $E(S\left(SC_{n}\right))$ in the random regular polygonal chains with *n* regular polygons, and we discuss the maximum and minimum of the $E(Gut\left(SC_{n}\right))$ and $E(S\left(SC_{n}\right))$. Meanwhile, we acquire the extremal values and the average values of the Gutman index and Schultz index among all regular polygonal chains with *n* regular polygons.

The random regular polygonal chain $SC_{n+1}$ with n+1 regular polygons consists of a new regular polygon $G_{n+1}$ attaching to the end of a regular polygonal chain $SC_{n}$ with *n* regular polygons; see Figure 1.

For n≥3, there are *k* ways to connect the terminal regular polygon $G_{n+1}$ with the front random regular polygonal chain $SC_{n}$. They can be described as $SC_{n+1}^{1},SC_{n+1}^{2},\cdots ,SC_{n+1}^{k-1}$ and $SC_{n+1}^{k}$, respectively; see Figure 2.

A random regular polygonal chain $SC_{n}$ with *n* regular polygons can be obtained by adding a regular polygon at the end of the chain step by step. At each step t(=3,4,…,n), a random connection is made from one of the following *k* possible cases:$SC_{t}\to SC_{t+1}^{1}$ with probability $p_{1}$,$SC_{t}\to SC_{t+1}^{2}$ with probability $p_{2}$,$SC_{t}\to SC_{t+1}^{3}$ with probability $p_{3}$,

⋯⋯$SC_{t}\to SC_{t+1}^{k}$ with probability $p_{k}=1-\left(p_{1}+p_{2}+p_{3}+\cdots +p_{k-1}\right)$,
where the *k* probabilities $p_{1},p_{2},\cdots ,p_{k-1}$ and $p_{k}$ are independent of the step parameter *t*. When $p_{1}=1,p_{2}=p_{3}=\cdots =p_{k}=0$, the meta-chain $M_{n}$ can be obtained. When $p_{2}=1,p_{1}=p_{3}=\cdots =p_{k}=0$, the orth-chain $O_{n}^{1}$ can be obtained. When $p_{3}=1,p_{1}=p_{2}=p_{4}=\cdots =p_{k}=0$, the orth-chain $O_{n}^{2}$ can be obtained. When $p_{k-1}=1$, the other k−1 probabilities reach 0, and we can obtain the orth-chain $O_{n}^{k-2}$. When $p_{k}=1,p_{1}=p_{2}=\cdots =p_{k-1}=0$, the para-chain $P_{n}$ can be obtained.

## 2. The Gutman Index of the Random Regular Polygonal Chain

For the random regular polygonal chain $SC_{n}$, we denote by $SC_{n+1}$ the graph acquired by connecting a new terminal regular polygon $G_{n+1}$ to $SC_{n}$, which is spanned by vertices $x_{1},x_{2},\cdots ,x_{2k}$ and $x_{1}$ is $u_{n}$ (see Figure 1). It is evident that, for all $v\in SC_{n}$, we have
(3)$$\begin{array}{c} d\left(v,x_{j}\right)=\left\{\begin{array}{c} d(v,u_{n})+j-1,\;\;\;\;\;\;\;1\leq j\leq k\\ d(v,u_{n})+2k+1-j,\;\;k+1\leq j\leq 2k. \end{array}\right.\; \end{array}$$
$$\sum_{v\in V(SC_{n})}d_{SC_{n+1}}\left(v\right)=4kn+2.$$

Meanwhile, we obtain that
(4)$$\begin{array}{c} \sum_{i=1}^{2k}d\left(x_{i}\right)d\left(x_{i},x_{j}\right)=\left\{\begin{array}{c} 2k^{2}+2\left(j-1\right),\;\;\;\;\;\;\;1\leq j\leq k\\ 2k^{2}+2\left(2k-j+1\right),\;\;k+1\leq j\leq 2k. \end{array}\right.\; \end{array}$$

**Theorem** **1.**
*The $E\left[Gut\left(SC_{n}\right)\right]\left(n\geq 1\right)$ of the random regular polygonal chain is*

$$\begin{array}{ccc} E[Gut\left(SC_{n}\right)] & = & \frac{8}{3}k^{2}n^{3}\left[k-\sum_{i=1}^{k-1}\left(k-i\right)p_{i}\right]+8k^{2}n^{2}\left[\sum_{i=1}^{k-1}\left(k-i\right)p_{i}\right]+\frac{4}{3}k^{2}n\left[k-4\sum_{i=1}^{k-1}\left(k-i\right)p_{i}\right]. \end{array}$$



**Proof.** Let $Gut\left(SC_{n}\right)=A_{1}+B_{1}+C_{1}$.
$$\begin{array}{ccc} A_{1} & = & \sum_{\left\{u,v\right\}\subseteq SC_{n}}d\left(u\right)d\left(v\right)d\left(u,v\right),\\ \;\;\; & = & \sum_{\left\{u,v\right\}\subseteq SC_{n}\setminus \left\{u_{n}\right\}}d\left(u\right)d\left(v\right)d\left(u,v\right)+\sum_{v\in SC_{n}\setminus \left\{u_{n}\right\}}d_{SC_{n+1}}\left(u_{n}\right)d\left(v\right)d\left(u_{n},v\right),\\ \;\;\; & = & \sum_{\left\{u,v\right\}\subseteq SC_{n}\setminus \left\{u_{n}\right\}}d\left(u\right)d\left(v\right)d\left(u,v\right)+\sum_{v\in SC_{n}\setminus \left\{u_{n}\right\}}\left[d_{SC_{n}}\left(u_{n}\right)+2\right]d\left(v\right)d\left(u_{n},v\right),\\ \;\;\; & = & Gut\left(SC_{n}\right)+2\sum_{v\in SC_{n}}d\left(v\right)d\left(u_{n},v\right).\\ B_{1} & = & \sum_{v\in SC_{n}\setminus \left\{u_{n}\right\}}\sum_{x_{i}\in G_{n+1}\setminus \left\{x_{1}\right\}}d\left(v\right)d\left(x_{i}\right)d\left(v,x_{i}\right),\\ \;\;\; & = & \sum_{v\in SC_{n}}\sum_{x_{i}\in G_{n}}d\left(v\right)d\left(x_{i}\right)d\left(v,x_{i}\right)-4\sum_{v\in SC_{n}}d\left(v\right)d\left(v,u_{n}\right)-4\sum_{v\in G_{n+1}}d\left(v\right)d\left(v,x_{1}\right),\\ \;\;\; & = & \sum_{v\in SC_{n}}d\left(v\right)\left\{4d\left(v,u_{n}\right)+4\left[d\left(v,u_{n}\right)+1\right]+4\left[d\left(v,u_{n}\right)+2\right]+\cdots +4\left[d\left(v,u_{n}\right)+k-1\right]+2\left[d\left(v,u_{n}\right)+k\right]\right\},\\ \;\;\; & & -4\sum_{v\in SC_{n}}d\left(v\right)d\left(v,u_{n}\right)-4\times 2k^{2},\\ \;\;\; & = & \left(4k-2\right)\sum_{v\in SC_{n}}d\left(v\right)d\left(v,u_{n}\right)+2k^{2}\left(4kn-2\right).\\ C_{1} & = & \sum_{\left\{x_{i},x_{j}\right\}\subseteq G_{n+1}}d\left(x_{i}\right)d\left(x_{j}\right)d\left(x_{i},x_{j}\right),\\ \;\;\; & = & \frac{1}{2}\sum_{i=1}^{2k}d\left(x_{i}\right)\left(\sum_{j=1}^{2k}d\left(x_{j}\right)d\left(x_{j},x_{i}\right)\right),\\ \;\;\; & = & 4k^{2}\left(k+1\right). \end{array}$$Then, $Gut\left(SC_{n+1}\right)=Gut\left(SC_{n}\right)+4k\sum_{v\in SC_{n}}d\left(v\right)d\left(v,u_{n}\right)+8k^{3}n+4k^{3}$.For convenience, we could denote the random variable $\sum_{v\in SC_{n}}d\left(v\right)d\left(v,u_{n}\right)$ by
$$U_{n}^{1}:=E\left(\sum_{v\in SC_{n}}d\left(v\right)d\left(v,u_{n}\right)\right).$$Therefore, we can obtain a recurrence relation as follows:
$$E\left(Gut\left(SC_{n+1}\right)\right)=E\left(Gut\left(SC_{n}\right)\right)+4kU_{n}^{1}+8k^{3}n+4k^{3}.$$By considering the following *k* possible conditions, it is easy to obtain $U_{n}^{1}$.**Condition** **1.**$SC_{n}⟶SC_{n+1}^{1}$*, then *$u_{n}$* would cover the vertex *$x_{2}$* or *$x_{2k}$*. Hence, *$\sum_{v\in V_{SC_{n}}}d\left(u_{n},v\right)$* can be represented as *$\sum_{v\in V_{SC_{n}}}d\left(x_{2},v\right)$* or *$\sum_{v\in V_{SC_{n}}}d\left(x_{2k},v\right)$* with probability *$p_{1}$.**Condition** **2.**$SC_{n}⟶SC_{n+1}^{2}$*, then *$u_{n}$* would cover the vertex *$x_{3}$* or *$x_{2k-1}$*. Hence, *$\sum_{v\in V_{SC_{n}}}d\left(u_{n},v\right)$* can be represented as *$\sum_{v\in V_{SC_{n}}}d\left(x_{3},v\right)$* or *$\sum_{v\in V_{SC_{n}}}d\left(x_{2k-1},v\right)$* with probability *$p_{2}$.

⋯⋯

**Condition** **3.**$SC_{n}⟶SC_{n+1}^{k-2}$*, then *$u_{n}$* would cover the vertex *$x_{k-1}$* or *$x_{k+3}$*. Hence, *$\sum_{v\in V_{SC_{n}}}d\left(u_{n},v\right)$* can be represented as *$\sum_{v\in V_{SC_{n}}}d\left(x_{k-1},v\right)$* or *$\sum_{v\in V_{SC_{n}}}d\left(x_{k+3},v\right)$* with probability *$p_{k-2}$.**Condition** **4.**$SC_{n}⟶SC_{n+1}^{k-1}$*, then *$u_{n}$* would cover the vertex *$x_{k}$* or *$x_{k+2}$*. Hence, *$\sum_{v\in V_{SC_{n}}}d\left(u_{n},v\right)$* can be represented as *$\sum_{v\in V_{SC_{n}}}d\left(x_{k},v\right)$* or *$\sum_{v\in V_{SC_{n}}}d\left(x_{k+2},v\right)$* with probability *$p_{k-1}$.**Condition** **5.**$SC_{n}⟶SC_{n+1}^{k}$*, then *$u_{n}$* would cover the vertex *$x_{k+1}$*. Hence, *$\sum_{v\in V_{SC_{n}}}d\left(u_{n},v\right)$* can be represented as *$\sum_{v\in V_{SC_{n}}}d\left(x_{k+1},v\right)$* with probability *$1-p_{1}-p_{2}-\cdots -p_{k-1}$.According to the *k* conditions, we have that
$$\begin{array}{ccc} U_{n}^{1} & = & p_{1}\sum_{v\in SC_{n}}d\left(v\right)d\left(v,x_{2}\right)+p_{2}\sum_{v\in SC_{n}}d\left(v\right)d\left(v,x_{3}\right)+\cdots +p_{k-1}\sum_{v\in SC_{n}}d\left(v\right)d\left(v,x_{k}\right)\\ \;\;\;\;\; & & +\left(1-p_{1}-p_{2}-\cdots -p_{k-1}\right)\sum_{v\in SPC_{n}}d\left(v\right)d\left(v,x_{k+1}\right),\\ \;\;\;\;\; & = & p_{1}\left[\sum_{v\in SC_{n-1}}d\left(v\right)d\left(v,u_{n-1}\right)+\sum_{v\in SC_{n-1}\setminus \left\{u_{n-1}\right\}}d\left(v\right)+2k^{2}+2\right]\\ \;\;\;\;\; & & +p_{2}\left[\sum_{v\in SC_{n-1}}d\left(v\right)d\left(v,u_{n-1}\right)+2\sum_{v\in SC_{n-1}\setminus \left\{u_{n-1}\right\}}d\left(v\right)+2k^{2}+4\right]+\cdots \\ \;\;\;\;\; & & +p_{k-1}\left[\sum_{v\in SC_{n-1}}d\left(v\right)d\left(v,u_{n-1}\right)+\left(k-1\right)\sum_{v\in SC_{n-1}\setminus \left\{u_{n-1}\right\}}d\left(v\right)+2k^{2}+2\left(k-1\right)\right]\\ \;\;\;\;\; & & +\left(1-p_{1}-p_{2}-\cdots -p_{k-1}\right)\left[\sum_{v\in SC_{n-1}}d\left(v\right)d\left(v,u_{n-1}\right)+k\sum_{v\in SC_{n-1}\setminus \left\{u_{n-1}\right\}}d\left(v\right)+2k^{2}+2k\right], \end{array}$$
$$\begin{array}{ccc} \;\;\;\;\; & = & p_{1}\left[\sum_{v\in SC_{n-1}}d\left(v\right)d\left(v,u_{n-1}\right)+4k\left(n-1\right)+2k^{2}\right]+p_{2}\left[\sum_{v\in SC_{n-1}}d\left(v\right)d\left(v,u_{n-1}\right)+2\cdot 4k\left(n-1\right)+2k^{2}\right]\\ \;\;\;\;\; & & +\cdots +p_{k-1}\left[\sum_{v\in SC_{n-1}}d\left(v\right)d\left(v,u_{n-1}\right)+\left(k-1\right)\cdot 4k\left(n-1\right)+2k^{2}\right]\\ \;\;\;\;\; & & +\left(1-p_{1}-p_{2}-\cdots -p_{k-1}\right)\left[\sum_{v\in SC_{n-1}}d\left(v\right)d\left(v,u_{n-1}\right)+k\cdot 4k\left(n-1\right)+2k^{2}\right],\\ \;\;\;\;\; & = & U_{n-1}^{1}+\left[4k^{2}-\left(k-1\right)\cdot 4kp_{1}-\left(k-2\right)\cdot 4kp_{2}-\cdots -4kp_{k-1}\right]n\\ \;\;\;\;\; & & +[-2k^{2}+\left(k-1\right)\cdot 4kp_{1}+\left(k-2\right)\cdot 4kp_{2}+\cdots +4kp_{k-1}],\\ \;\;\;\;\; & = & U_{n-1}^{1}+\left[4k^{2}-4k\sum_{i=1}^{k-1}\left(k-i\right)p_{i}\right]n+\left[-2k^{2}+4k\sum_{i=1}^{k-1}\left(k-i\right)p_{i}\right]. \end{array}$$Moreover, the original value is $U_{1}^{1}=\sum_{v\in SC_{1}}d\left(v\right)d\left(v,u_{1}\right)=2k^{2}$. Thus,
$$U_{n}^{1}=\left[2k^{2}-2k\sum_{i=1}^{k-1}\left(k-i\right)p_{i}\right]n^{2}+\left[2k\sum_{i=1}^{k-1}\left(k-i\right)p_{i}\right]n.$$Due to
$$\begin{array}{ccc} E(Gut\left(SC_{n+1}\right)) & = & E\left(Gut\left(SC_{n}\right)\right)+4kU_{n}^{1}+8k^{3}n+4k^{3},\\ \;\;\;\;\;\;\;\; & = & E\left(Gut\left(SPC_{n}\right)\right)+8k^{2}n^{2}\left[k-\left(k-1\right)p_{1}-\left(k-2\right)p_{2}-\cdots -p_{k-1}\right],\\ \;\;\;\;\;\;\;\; & & +8k^{2}n\left[k+\left(k-1\right)p_{1}+\left(k-2\right)p_{2}+\cdots +p_{k-1}\right]+4k^{3},\\ \;\;\;\;\;\;\;\; & = & E\left(Gut\left(SPC_{n}\right)\right)+8k^{2}n^{2}\left[k-\sum_{i=1}^{k-1}\left(k-i\right)p_{i}\right]+8k^{2}n\left[k+\sum_{i=1}^{k-1}\left(k-i\right)p_{i}\right]+4k^{3}. \end{array}$$From the original value, $E\left(Gut\left(SC_{1}\right)\right)=2\times 2\times 2k\times \frac{1}{2}\times \left[1\times 2+2\times 2+\cdots +2\times \left(k-1\right)+k\right]=4k^{3}$, and the above recurrence relation, we may calculate that
$$\begin{array}{ccc} \;E(Gut\left(SC_{n}\right)) & = & E\left(Gut\left(SC_{n}\right)\right)=\frac{8}{3}k^{2}n^{3}\left[k-\left(k-1\right)p_{1}-\left(k-2\right)p_{2}-\cdots -p_{k-1}\right]\\ \;\;\;\;\;\;\; & & +8k^{2}n^{2}\left[\left(k-1\right)p_{1}+\left(k-2\right)p_{2}+\cdots +p_{k-1}\right]\\ \;\;\;\;\;\;\; & & +\frac{4}{3}k^{2}n\left[k-4\left(k-1\right)p_{1}-4\left(k-2\right)p_{2}-\cdots -4p_{k-1}\right],\\ \;\;\;\;\;\;\; & = & \frac{8}{3}k^{2}n^{3}\left[k-\sum_{i=1}^{k-1}\left(k-i\right)p_{i}\right]+8k^{2}n^{2}\left[\sum_{i=1}^{k-1}\left(k-i\right)p_{i}\right]+\frac{4}{3}k^{2}n\left[k-4\sum_{i=1}^{k-1}\left(k-i\right)p_{i}\right]. \end{array}$$The proof is complete. □

According to the proof of Theorem 1, if $p_{1}=1$, each of the other k−1 probabilities is equal to 0, and then we have $SC_{n}≅M_{n}$. Likewise, if $p_{2}=1$, each of the other k−1 probabilities is equal to 0, and then $SC_{n}≅O_{n}^{1}$; if $p_{3}=1$, each of the other k−1 probabilities is equal to 0, and then $SC_{n}≅O_{n}^{2}$ and so on; if $p_{k-1}=1$, each of the other k−1 probabilities is equal to 0, and then $SC_{n}≅O_{n}^{k-2}$; if $p_{k}=1$, each of the other k−1 probabilities is equal to 0, and then $SC_{n}≅P_{n}$.

**Corollary** **1.**
*The Gutman indices of $M_{n}$, $O_{n}^{1},O_{n}^{2},\cdots ,O_{n}^{k-2}$, $P_{n}$ are*

$$Gut\left(M_{n}\right)=\frac{8}{3}k^{2}n^{3}+8k^{2}\left(k-1\right)n^{2}+\frac{4}{3}k^{2}\left(4-3k\right)n;$$


$$Gut\left(O_{n}^{1}\right)=2\cdot \frac{8}{3}k^{2}n^{3}+8k^{2}\left(k-2\right)n^{2}+\frac{4}{3}\left(8-3k\right)k^{2}n;$$


$$Gut\left(O_{n}^{2}\right)=3\cdot \frac{8}{3}k^{2}n^{3}+8k^{2}\left(k-3\right)n^{2}+\frac{4}{3}\left(12-3k\right)k^{2}n;$$


⋯⋯


$$Gut\left(O_{n}^{k-2}\right)=\left(k-1\right)\cdot \frac{8}{3}k^{2}n^{3}+8k^{2}n^{2}+\frac{4}{3}\left[\left(k-1\right)\cdot 4-3k\right]k^{2}n;$$


$$Gut\left(P_{n}\right)=\frac{8}{3}k^{3}n^{3}+\frac{4}{3}k^{3}n.$$



**Corollary** **2.**
*Among all polygonal chains with n(n⩾3) 2k polygons, $P_{n}$ realizes the maximum of $E(Gut\left(SC_{n}\right))$ and $M_{n}$ realizes the minimum of $E(Gut\left(SC_{n}\right))$.*


**Proof.** By Theorem 1, we have
$$\begin{array}{ccc} f_{1} & = & E(Gut\left(SC_{n}\right)),\\ \;\;\; & = & \left[-\frac{8}{3}k^{2}\left(k-1\right)n^{3}+8k^{2}\left(k-1\right)n^{2}-\frac{16}{3}k^{2}\left(k-1\right)n\right]p_{1}\\ \;\;\; & & +\left[-\frac{8}{3}k^{2}\left(k-2\right)n^{3}+8k^{2}\left(k-2\right)n^{2}-\frac{16}{3}k^{2}\left(k-2\right)n\right]p_{2}+\cdots \\ \;\;\; & & +\left[-\frac{8}{3}k^{2}n^{3}+8k^{2}n^{2}-\frac{16}{3}k^{2}n\right]p_{k-1}+\left(\frac{8}{3}k^{3}n^{3}+\frac{4}{3}k^{3}n\right). \end{array}$$
as n≥3, we have that
$$\frac{\partial f_{1}}{\partial p_{1}}=-\frac{8}{3}k^{2}\left(k-1\right)n^{3}+8k^{2}\left(k-1\right)n^{2}-\frac{16}{3}k^{2}\left(k-1\right)n<0;$$
$$\frac{\partial f_{1}}{\partial p_{2}}=-\frac{8}{3}k^{2}\left(k-2\right)n^{3}+8k^{2}\left(k-2\right)n^{2}-\frac{16}{3}k^{2}\left(k-2\right)n<0;$$
⋯⋯
$$\frac{\partial f_{1}}{\partial p_{k-1}}=-\frac{8}{3}k^{2}n^{3}+8k^{2}n^{2}-\frac{16}{3}k^{2}n<0.$$When $\left(p_{1},p_{2},\cdots ,p_{k-1},p_{k}\right)=\left(0,0,\cdots ,0,1\right)$, $P_{n}$ realizes the maximum of $E[Gut\left(SC_{n}\right)]$, thus, $SC_{n}≅P_{n}$. If $p_{k}=0$,$p_{1}+p_{2}+\cdots +p_{k-1}=1$, we can translate this into $p_{k-1}=1-p_{1}-p_{2}-\cdots -p_{k-2}\;\left(0\leq p_{l}\leq 1,l\in \left[1,k-2\right]\right)$, then
$$\begin{array}{ccc} E[Gut\left(SC_{n}\right)] & = & \left[-\frac{8}{3}k^{2}\left(k-1\right)n^{3}+8k^{2}\left(k-1\right)n^{2}-\frac{16}{3}k^{2}\left(k-1\right)n\right]p_{1}\\ \;\;\;\;\;\;\; & & +\left[-\frac{8}{3}k^{2}\left(k-2\right)n^{3}+8k^{2}\left(k-2\right)n^{2}-\frac{16}{3}k^{2}\left(k-2\right)n\right]p_{2}+\cdots \\ \;\;\;\;\;\;\; & & +\left[-\frac{8}{3}k^{2}n^{3}+8k^{2}n^{2}-\frac{16}{3}k^{2}n\right]\left(1-p_{1}-p_{2}-\cdots -p_{k-2}\right)\\ \;\;\;\;\;\;\; & & +(\frac{8}{3}k^{3}n^{3}+\frac{4}{3}k^{3}n). \end{array}$$Thus,
$$\begin{array}{cc} \frac{\partial E(Gut\left(SC_{n}\right))}{\partial p_{1}} & =-\frac{8}{3}k^{2}\left(k-2\right)n^{3}+8k^{2}\left(k-2\right)n^{2}-\frac{16}{3}k^{2}\left(k-2\right)n<0,\\ \frac{\partial E(Gut\left(SC_{n}\right))}{\partial p_{2}} & =-\frac{8}{3}k^{2}\left(k-3\right)n^{3}+8k^{2}\left(k-3\right)n^{2}-\frac{16}{3}k^{2}\left(k-3\right)n<0, \end{array}$$
⋯⋯
$$\begin{array}{cc} \frac{\partial E(Gut\left(SC_{n}\right))}{\partial p_{k-2}} & =-\frac{8}{3}k^{2}n^{3}+8k^{2}n^{2}-\frac{16}{3}k^{2}n<0. \end{array}$$However, $p_{1}=p_{2}=\cdots =p_{k-2}=0$ (i.e., $p_{k-1}=1$), and we cannot acquire the minimum value of $E(Gut\left(SC_{n}\right))$. Thus, we consider $p_{1}+p_{2}+\cdots +p_{k-2}=1$, then $p_{k-2}=1-p_{1}-p_{2}-\cdots -p_{k-3}\;\left(0\leq p_{l}\leq 1,l\in \left[1,k-3\right]\right)$.
$$\begin{array}{ccc} E[Gut\left(SC_{n}\right)] & = & \left[-\frac{8}{3}k^{2}\left(k-1\right)n^{3}+8k^{2}\left(k-1\right)n^{2}-\frac{16}{3}k^{2}\left(k-1\right)n\right]p_{1}\\ \;\;\;\;\;\;\; & & +\left[-\frac{8}{3}k^{2}\left(k-2\right)n^{3}+8k^{2}\left(k-2\right)n^{2}-\frac{16}{3}k^{2}\left(k-2\right)n\right]p_{2}+\cdots \\ \;\;\;\;\;\;\; & & +\left[-2\cdot \frac{8}{3}k^{2}n^{3}+2\cdot 8k^{2}n^{2}-2\cdot \frac{16}{3}k^{2}n\right]\left(1-p_{1}-p_{2}-\cdots -p_{k-3}\right)\\ \;\;\;\;\;\;\; & & +(\frac{8}{3}k^{3}n^{3}+\frac{4}{3}k^{3}n). \end{array}$$Thus,
$$\begin{array}{cc} \frac{\partial E(Gut\left(SC_{n}\right))}{\partial p_{1}} & =-\frac{8}{3}k^{2}\left(k-3\right)n^{3}+8k^{2}\left(k-3\right)n^{2}-\frac{16}{3}k^{2}\left(k-3\right)n<0,\\ \frac{\partial E(Gut\left(SC_{n}\right))}{\partial p_{2}} & =-\frac{8}{3}k^{2}\left(k-4\right)n^{3}+8k^{2}\left(k-4\right)n^{2}-\frac{16}{3}k^{2}\left(k-4\right)n<0, \end{array}$$
⋯⋯
$$\begin{array}{cc} \frac{\partial E(Gut\left(SC_{n}\right))}{\partial p_{k-3}} & =-\frac{8}{3}k^{2}n^{3}+8k^{2}n^{2}-\frac{16}{3}k^{2}n<0. \end{array}$$However, $p_{1}=p_{2}=\cdots =p_{k-3}=0$ (i.e., $p_{k-2}=1$), and $E(Gut\left(SC_{n}\right))$ cannot attain the minimum value. By this analogy, if $p_{1}+p_{2}=1$, let $p_{1}=1-p_{2}\left(0\leq p_{2}\leq 1\right)$.
$$\begin{array}{ccc} E[Gut\left(SC_{n}\right)] & = & \left[-\frac{8}{3}k^{2}\left(k-1\right)n^{3}+8k^{2}\left(k-1\right)n^{2}-\frac{16}{3}k^{2}\left(k-1\right)n\right]\left(1-p_{2}\right)\\ \;\;\;\;\;\;\; & & +\left[-\frac{8}{3}k^{2}\left(k-2\right)n^{3}+8k^{2}\left(k-2\right)n^{2}-\frac{16}{3}k^{2}\left(k-2\right)n\right]p_{2}\\ \;\;\;\;\;\;\; & & +(\frac{8}{3}k^{3}n^{3}+\frac{4}{3}k^{3}n). \end{array}$$Thus,
$$\frac{\partial E(Gut\left(SC_{n}\right))}{\partial p_{2}}=\frac{8}{3}k^{2}n^{3}-8k^{2}n^{2}+\frac{16}{3}k^{2}n>0.$$Therefore, $E(Gut\left(SC_{n}\right))$ achieves the minimum value, if $p_{1}=1,p_{2}=0$, which will be $SC_{n}≅M_{n}$. This completes the proof. □

## 3. The Schultz Index of the Random Regular Polygonal Chain

The Schultz index of a random polygonal chain $SC_{n}$ is a random variable. We calculate the expected value of $S(SC_{n})$ as follows. Denote by $E(S\left(SC_{n}\right))$ the expected value of the Schultz index of the random polygonal chain $SC_{n}$.

**Theorem** **2.**
*The $E\left[S\left(SC_{n}\right)\right]\left(n\geq 1\right)$ of the random regular polygonal chain is*

$$\begin{array}{ccc} E[S\left(SC_{n}\right)] & = & \frac{4k(2k-1)}{3}n^{3}\left[k-\sum_{i=1}^{k-1}\left(k-i\right)p_{i}\right]+2kn^{2}\left[k+2\left(2k-1\right)\sum_{i=1}^{k-1}\left(k-i\right)p_{i}\right]\\ \;\;\;\;\;\; & & +\frac{2k(2k-1)}{3}n\left[k-4\sum_{i=1}^{k-1}\left(k-i\right)p_{i}\right]. \end{array}$$



**Proof.** Let $S\left(SC_{n}\right)=A_{2}+B_{2}+C_{2}$.
$$\begin{array}{ccc} A_{2} & = & \sum_{\left\{u,v\right\}\subseteq SC_{n}}\left[d\left(u\right)+d\left(v\right)\right]d\left(u,v\right),\\ \;\;\; & = & \sum_{\left\{u,v\right\}\subseteq SC_{n}\setminus \left\{u_{n}\right\}}\left[d\left(u\right)+d\left(v\right)\right]d\left(u,v\right)+\sum_{v\in SC_{n}\setminus \left\{u_{n}\right\}}\left[d_{SC_{n+1}}\left(u_{n}\right)+d\left(v\right)\right]d\left(u_{n},v\right),\\ \;\;\; & = & \sum_{\left\{u,v\right\}\subseteq SC_{n}\setminus \left\{u_{n}\right\}}\left[d\left(u\right)+d\left(v\right)\right]d\left(u,v\right)+\sum_{v\in SC_{n}\setminus \left\{u_{n}\right\}}\left[d_{SC_{n}}\left(u_{n}\right)+2+d\left(v\right)\right]d\left(u_{n},v\right),\\ \;\;\; & = & S\left(SC_{n}\right)+2\sum_{v\in SC_{n}}d\left(u_{n},v\right),\\ \;\;\; & = & S\left(SC_{n}\right)+2d\left(u_{n}|SC_{n}\right).\\ B_{2} & = & \sum_{v\in SC_{n}\setminus \left\{u_{n}\right\}}\sum_{x_{i}\in G_{n+1}\setminus \left\{x_{1}\right\}}\left[d\left(v\right)+d\left(x_{i}\right)\right]d\left(v,x_{i}\right),\\ \;\;\; & = & \sum_{v\in SC_{n}}\sum_{x_{i}\in G_{n}}\left[d\left(v\right)+d\left(x_{i}\right)\right]d\left(v,x_{i}\right)-\sum_{v\in SC_{n}}\left[d\left(v\right)+4\right]d\left(v,u_{n}\right)-\sum_{v\in G_{n+1}}\left[d\left(v\right)+4\right]d\left(v,x_{1}\right),\\ \;\;\; & = & \sum_{v\in SC_{n}}d\left(v\right)\left\{d\left(v,u_{n}\right)+2\left[d\left(v,u_{n}\right)+1\right]+2\left[d\left(v,u_{n}\right)+2\right]+\cdots +2\left[d\left(v,u_{n}\right)+k-1\right]+\left[d\left(v,u_{n}\right)+k\right]\right\}\\ \;\;\; & & +\sum_{v\in SC_{n}}\left\{4d\left(v,u_{n}\right)+4\left[d\left(v,u_{n}\right)+1\right]+4\left[d\left(v,u_{n}\right)+2\right]+\cdots +4\left[d\left(v,u_{n}\right)+k-1\right]+2\left[d\left(v,u_{n}\right)+k\right]\right\}\\ \;\;\; & & -\sum_{v\in SC_{n}}d\left(v\right)d\left(v,u_{n}\right)-4\sum_{v\in SC_{n}}d\left(v,u_{n}\right)-6k^{2},\\ \;\;\; & = & \left(2k-1\right)\sum_{v\in SC_{n}}d\left(v\right)d\left(v,u_{n}\right)+\left(4k-2\right)d\left(u_{n}|SC_{n}\right)+\left(8k^{3}-2k^{2}\right)n-2k^{2}.\\ C_{2} & = & \sum_{\left\{x_{i},x_{j}\right\}\subseteq G_{n+1}}\left[d\left(x_{i}\right)+d\left(x_{j}\right)\right]d\left(x_{i},x_{j}\right),\\ \;\;\; & = & \sum_{i=1}^{2k}d\left(x_{i}\right)\sum_{j=1}^{2k}d\left(x_{j},x_{i}\right),\\ \;\;\; & = & 4k^{3}+2k^{2}. \end{array}$$Then, $S\left(SC_{n+1}\right)=S\left(SC_{n}\right)+\left(2k-1\right)\sum_{v\in SC_{n}}d\left(v\right)d\left(v,u_{n}\right)+4kd\left(u_{n}|SPC_{n}\right)+\left(8k^{3}-2k^{2}\right)n+4k^{3}$.For convenience, we could denote the random variable $d(u_{n}|SC_{n})$ by
$$U_{n}^{2}:=E\left(d\left(u_{n}|SC_{n}\right)\right).$$Therefore, we can obtain a recurrence relation as follows:
$$E\left(S\left(SC_{n+1}\right)\right)=E\left(S\left(SC_{n}\right)\right)+7U_{n}^{1}+16U_{n}^{2}+480n+256$$By considering the following *k* possible conditions, it is easy to obtain $U_{n}^{2}$.**Condition** **6.**$SC_{n}⟶SC_{n+1}^{1}$*, then *$u_{n}$* would cover the vertex *$x_{2}$* or *$x_{2k}$*. Hence, *$d(u_{n}|SC_{n})$* can be represented as *$d(x_{2}|SC_{n})$* or *$d(x_{2k}|SC_{n})$* with probability *$p_{1}$.**Condition** **7.**$SC_{n}⟶SC_{n+1}^{2}$*, then *$u_{n}$* would cover the vertex *$x_{3}$* or *$x_{2k-1}$*. Hence, *$d(u_{n}|SC_{n})$* can be represented as *$d(x_{3}|SC_{n})$* or *$d(x_{2k-1}|SC_{n})$* with probability *$p_{2}$.

⋯⋯

**Condition** **8.**$SC_{n}⟶SC_{n+1}^{k-2}$*, then *$u_{n}$* would cover the vertex *$x_{k-1}$* or *$x_{k+3}$*. Hence, *$d(u_{n}|SC_{n})$* can be represented as *$d(x_{k-1}|SC_{n})$* or *$d(x_{k+3}|SC_{n})$* with probability *$p_{k-2}$.**Condition** **9.**$SC_{n}⟶SC_{n+1}^{k-1}$*, then *$u_{n}$* would cover the vertex *$x_{k}$* or *$x_{k+2}$*. Hence, *$d(u_{n}|SC_{n})$* can be represented as *$d(x_{k}|SC_{n})$* or *$d(x_{k+2}|SC_{n})$* with probability *$p_{k-1}$.**Condition** **10.**$SC_{n}⟶SC_{n+1}^{k}$*, then *$u_{n}$* would cover the vertex *$x_{k+1}$*. Hence, *$d(u_{n}|SC_{n})$* can be represented as *$d(x_{k+1}|SC_{n})$* with probability *$1-p_{1}-p_{2}-\cdots -p_{k-1}$.According to the *k* conditions, we have that
$$\begin{array}{ccc} U_{n}^{2} & = & p_{1}d\left(x_{2}|SC_{n}\right)+p_{2}d\left(x_{3}|SC_{n}\right)+\cdots +p_{k-1}d\left(x_{k}|SC_{n}\right)+\left(1-p_{1}-p_{2}-\cdots -p_{k-1}\right)d\left(x_{k+1}|SC_{n}\right),\\ \;\;\;\;\; & = & p_{1}\left[d\left(u_{n-1}|SC_{n}\right)+\left(2k-1\right)\left(n-1\right)+k^{2}\right]+p_{2}\left[d\left(u_{n-1}|SC_{n}\right)+2\left(2k-1\right)\left(n-1\right)+k^{2}\right]+\cdots \\ \;\;\;\;\; & & +p_{k-1}\left[d\left(u_{n-1}|SC_{n}\right)+\left(k-1\right)\left(2k-1\right)\left(n-1\right)+k^{2}\right]\\ \;\;\;\;\; & & +\left(1-p_{1}-p_{2}-\cdots -p_{k-1}\right)\left[d\left(u_{n-1}|SC_{n}\right)+k\left(2k-1\right)\left(n-1\right)+k^{2}\right],\\ \;\;\;\;\; & = & U_{n-1}^{2}+\left[k\left(2k-1\right)-\left(k-1\right)\left(2k-1\right)p_{1}-\left(k-2\right)\left(2k-1\right)p_{2}-\cdots -\left(2k-1\right)p_{k-1}\right]n\\ \;\;\;\;\; & & +[\left(k-k^{2}\right)+\left(k-1\right)\left(2k-1\right)p_{1}+\left(k-2\right)\left(2k-1\right)p_{2}+\cdots +\left(2k-1\right)p_{k-1}],\\ \;\;\;\;\; & = & U_{n-1}^{2}+\left[k\left(2k-1\right)-\left(2k-1\right)\sum_{i=1}^{k-1}\left(k-i\right)p_{i}\right]n+\left[\left(k-k^{2}\right)+\left(2k-1\right)\sum_{i=1}^{k-1}\left(k-i\right)p_{i}\right]. \end{array}$$Moreover, the original value is $U_{1}^{2}=d\left(u_{1}|SC_{1}\right)=k^{2}$. Thus,
$$\begin{array}{ccc} U_{n}^{2} & = & \frac{1}{2}\left[k\left(2k-1\right)-\left(k-1\right)\left(2k-1\right)p_{1}-\left(k-2\right)\left(2k-1\right)p_{2}-\cdots -\left(2k-1\right)p_{k-1}\right]n^{2}\\ \;\;\;\;\; & = & +\frac{1}{2}\left[k+\left(k-1\right)\left(2k-1\right)p_{1}+\left(k-2\right)\left(2k-1\right)p_{2}+\cdots +\left(2k-1\right)p_{k-1}\right]n,\\ \;\;\;\;\; & = & \frac{1}{2}\left[k\left(2k-1\right)-\left(2k-1\right)\sum_{i-1}^{k-1}\left(k-i\right)p_{i}\right]n^{2}+\frac{1}{2}\left[k+\left(2k-1\right)\sum_{i=1}^{k-1}\left(k-i\right)p_{i}\right]n. \end{array}$$Due to
$$\begin{array}{ccc} E(S\left(SC_{n+1}\right)) & = & E\left(S\left(SC_{n}\right)\right)+\left(2k-1\right)U_{n}^{1}+4kU_{n}^{2}+\left(8k^{3}-2k^{2}\right)n+4k^{3},\\ \;\;\;\;\;\;\; & = & E\left(S\left(SC_{n}\right)\right)+4k\left(2k-1\right)n^{2}\left[k-\left(k-1\right)p_{1}-\left(k-2\right)p_{2}-\cdots -p_{k-1}\right]\\ \;\;\;\;\;\;\; & & +4kn\left[2k^{2}+\left(k-1\right)\left(2k-1\right)p_{1}+\left(k-2\right)\left(2k-1\right)p_{2}+\cdots +\left(2k-1\right)p_{k-1}\right]+4k^{3},\\ \;\;\;\;\;\;\; & = & E\left(S\left(SC_{n}\right)\right)+4k\left(2k-1\right)n^{2}\left[k-\sum_{i=1}^{k-1}\left(k-i\right)p_{i}\right]+4kn\left[2k^{2}+\left(2k-1\right)\sum_{i=1}^{k-1}\left(k-i\right)p_{i}\right]. \end{array}$$From the original value, $E\left(S\left(SC_{1}\right)\right)=k^{2}\times 2k\times 2=4k^{3}$, and the above recurrence relation, we may calculate that
$$\begin{array}{ccc} E(S\left(SC_{n}\right)) & = & \frac{4k(2k-1)}{3}n^{3}\left[k-\left(k-1\right)p_{1}-\left(k-2\right)p_{2}-\cdots -p_{k-1}\right]\\ \;\;\;\;\;\; & & +2kn^{2}\left[k+2\left(k-1\right)\left(2k-1\right)p_{1}+2\left(k-2\right)\left(2k-1\right)p_{2}+\cdots +2\left(2k-1\right)p_{k-1}\right]\\ \;\;\;\;\;\; & & +\frac{2k(2k-1)}{3}n\left[k-4\left(k-1\right)p_{1}-4\left(k-2\right)p_{2}-\cdots -4p_{k-1}\right],\\ \;\;\;\;\;\; & = & \frac{4k(2k-1)}{3}n^{3}\left[k-\sum_{i=1}^{k-1}\left(k-i\right)p_{i}\right]+2kn^{2}\left[k+2\left(2k-1\right)\sum_{i=1}^{k-1}\left(k-i\right)p_{i}\right]\\ \;\;\;\;\;\; & & +\frac{2k(2k-1)}{3}n\left[k-4\sum_{i=1}^{k-1}\left(k-i\right)p_{i}\right]. \end{array}$$The proof is complete. □

According to the proof of Theorem 2, if $p_{1}=1$, each of the other k−1 probabilities is equal to 0, and then we have $SC_{n}≅M_{n}$. Likewise, if $p_{2}=1$, each of the other k−1 probabilities is equal to 0, and then $SC_{n}≅O_{n}^{1}$; if $p_{3}=1$, each of the other k−1 probabilities is equal to 0, and then $SC_{n}≅O_{n}^{2}$ and so on; if $p_{k-1}=1$, each of the other k−1 probabilities is equal to 0, and then $SC_{n}≅O_{n}^{k-2}$; if $p_{k}=1$, each of the other k−1 probabilities is equal to 0, and then $SC_{n}≅P_{n}$.

**Corollary** **3.**
*The Schultz indices of $M_{n}$, $O_{n}^{1},O_{n}^{2},\cdots ,O_{n}^{k-2}$, $P_{n}$ are*

$$S\left(M_{n}\right)=\frac{4k(2k-1)}{3}n^{3}+2k\left(2k-1\right)\left(k-1\right)n^{2}+\frac{2k(2k-1)(4-3k)}{3}n;$$


$$S\left(O_{n}^{1}\right)=2\cdot \frac{4k(2k-1)}{3}n^{3}+2k\left[2\left(2k-1\right)\left(k-2\right)+k\right]n^{2}+\frac{2k(2k-1)}{3}\left[k+4\left(2-k\right)\right]n;$$


$$S\left(O_{n}^{2}\right)=3\cdot \frac{4k(2k-1)}{3}n^{3}+2k\left[2\left(2k-1\right)\left(k-3\right)+k\right]n^{2}+\frac{2k(2k-1)}{3}\left[k+4\left(3-k\right)\right]n;$$


⋯⋯


$$S\left(O_{n}^{k-2}\right)=\left(k-1\right)\cdot \frac{4k(2k-1)}{3}n^{3}+2k\left[2\left(2k-1\right)+k\right]n^{2}+\frac{2k(2k-1)}{3}\left[k+4\left(-1\right)\right]n;$$


$$S\left(P_{n}\right)=k\cdot \frac{4k(2k-1)}{3}n^{3}+2k\cdot kn^{2}+k\cdot \frac{2k(2k-1)}{3}n.$$



**Corollary** **4.**
*Among all polygonal chains with n(n⩾3) 2k polygons, $P_{n}$ realizes the maximum of $E(S\left(SC_{n}\right))$ and the meta-chain $M_{n}$ realizes that of the minimum.*


**Proof.** By Theorem 2, we can deduce
$$\begin{array}{ccc} f_{2} & = & E(S\left(SC_{n}\right)),\\ \;\;\; & = & \left(2k-1\right)\left[-\frac{4k(k-1)}{3}n^{3}+4k\left(k-1\right)n^{2}-\frac{8k(k-1)}{3}n\right]p_{1}\\ \;\;\; & & +\left(2k-1\right)\left[-\frac{4k(k-2)}{3}n^{3}+4k\left(k-2\right)n^{2}-\frac{8k(k-2)}{3}n\right]p_{2}+\cdots \\ \;\;\; & & +\left(2k-1\right)\left[-\frac{4k}{3}n^{3}+4kn^{2}-\frac{8k}{3}n\right]p_{k-1}+\left[\frac{4k(2k-1)}{3}kn^{3}+2k\cdot kn^{2}+\frac{2k^{2}\left(2k-1\right)}{3}n\right]. \end{array}$$
as n≥3, we have that
$$\frac{\partial f_{2}}{\partial p_{1}}=\left(2k-1\right)\left[-\frac{4k(k-1)}{3}k^{2}n^{3}+4k\left(k-1\right)n^{2}-\frac{8k(k-1)}{3}n\right]<0;$$
$$\frac{\partial f_{2}}{\partial p_{2}}=\left(2k-1\right)\left[-\frac{4k(k-2)}{3}k^{2}n^{3}+4k\left(k-2\right)n^{2}-\frac{8k(k-2)}{3}n\right]<0;$$
⋯⋯
$$\frac{\partial f_{2}}{\partial p_{k-1}}=\left(2k-1\right)\left[-\frac{4k}{3}k^{2}n^{3}+4kn^{2}-\frac{8k}{3}n\right]<0.$$When $\left(p_{1},p_{2},\cdots ,p_{k-1},p_{k}\right)=\left(0,0,\cdots ,0,1\right)$, $P_{n}$ realizes the maximum of $E[S\left(SC_{n}\right)]$; thus, $SC_{n}≅P_{n}$. If $p_{k}=0$,$p_{1}+p_{2}+\cdots +p_{k-1}=1$, we can translate this into $p_{k-1}=1-p_{1}-p_{2}-\cdots -p_{k-2}\;\left(0\leq p_{l}\leq 1,l\in \left[1,k-2\right]\right)$, then
$$\begin{array}{ccc} E[S\left(SC_{n}\right)] & = & \left(2k-1\right)\left[-\frac{4k(k-1)}{3}n^{3}+4k\left(k-1\right)n^{2}-\frac{8k(k-1)}{3}n\right]p_{1}\\ \;\;\;\;\;\; & & +\left(2k-1\right)\left[-\frac{4k(k-2)}{3}n^{3}+4k\left(k-2\right)n^{2}-\frac{8k(k-2)}{3}n\right]p_{2}+\cdots \\ \;\;\;\;\;\; & & +\left(2k-1\right)\left[-\frac{4k}{3}n^{3}+4kn^{2}-\frac{8k}{3}n\right]\left(1-p_{1}-p_{2}-\cdots -p_{k-2}\right)\\ \;\;\;\;\;\; & & +[\frac{4k(2k-1)}{3}kn^{3}+2k\cdot kn^{2}+\frac{2k^{2}\left(2k-1\right)}{3}n]. \end{array}$$Thus,
$$\begin{array}{cc} \frac{\partial E(S\left(SC_{n}\right))}{\partial p_{1}} & =\left(2k-1\right)[-\frac{4k(k-2)}{3}n^{3}+4k\left(k-2\right)n^{2}-\frac{8k(k-2)}{3}n]<0,\\ \frac{\partial E(S\left(SC_{n}\right))}{\partial p_{2}} & =\left(2k-1\right)[-\frac{4k(k-3)}{3}n^{3}+4k\left(k-3\right)n^{2}-\frac{8k(k-3)}{3}n]<0, \end{array}$$
⋯⋯
$$\begin{array}{cc} \frac{\partial E(S\left(SC_{n}\right))}{\partial p_{k-2}} & =\left(2k-1\right)\left[-\frac{4k}{3}n^{3}+4kn^{2}-\frac{8k}{3}n\right]<0. \end{array}$$However, $p_{1}=p_{2}=\cdots =p_{k-2}=0$ (i.e., $p_{k-1}=1$), and we cannot acquire the minimum value of $E(S\left(SC_{n}\right))$. Thus, consider $p_{1}+p_{2}+\cdots +p_{k-2}=1$, then $p_{k-2}=1-p_{1}-p_{2}-\cdots -p_{k-3}\;\left(0\leq p_{l}\leq 1,l\in \left[1,k-3\right]\right)$.
$$\begin{array}{ccc} E[S\left(SC_{n}\right)] & = & \left(2k-1\right)\left[-\frac{4k(k-1)}{3}n^{3}+4k\left(k-1\right)n^{2}-\frac{8k(k-1)}{3}n\right]p_{1}\\ \;\;\;\;\;\; & & +\left(2k-1\right)\left[-\frac{4k(k-2)}{3}n^{3}+4k\left(k-2\right)n^{2}-\frac{8k(k-2)}{3}n\right]p_{2}+\cdots \\ \;\;\;\;\;\; & & +\left(2k-1\right)\left[-2\cdot \frac{4k}{3}n^{3}+2\cdot 4kn^{2}-2\cdot \frac{8k}{3}n\right]\left(1-p_{1}-p_{2}-\cdots -p_{k-3}\right)\\ \;\;\;\;\;\; & & +[\frac{4k(2k-1)}{3}n^{3}+2k\cdot kn^{2}+\frac{2k(2k-1)}{3}n]\\ \;\;\;\;\;\; & & +[\frac{4k(2k-1)}{3}kn^{3}+2k\cdot kn^{2}+\frac{2k^{2}\left(2k-1\right)}{3}n]. \end{array}$$Thus,
$$\begin{array}{cc} \frac{\partial E(S\left(SC_{n}\right))}{\partial p_{1}} & =\left(2k-1\right)[-\frac{4k(k-3)}{3}n^{3}+4k\left(k-3\right)n^{2}-\frac{8k(k-3)}{3}n]<0,\\ \frac{\partial E(S\left(SC_{n}\right))}{\partial p_{2}} & =\left(2k-1\right)[-\frac{4k(k-4)}{3}n^{3}+4k\left(k-4\right)n^{2}-\frac{8k(k-4)}{3}n]<0, \end{array}$$
⋯⋯
$$\begin{array}{cc} \frac{\partial E(S\left(SC_{n}\right))}{\partial p_{k-3}} & =\left(2k-1\right)\left[-\frac{4k}{3}n^{3}+4kn^{2}-\frac{8k}{3}n\right]<0. \end{array}$$However, $p_{1}=p_{2}=\cdots =p_{k-3}=0$ (i.e., $p_{k-2}=1$), and $E(S\left(SC_{n}\right))$ cannot attain the minimum value. By this analogy, if $p_{1}+p_{2}=1$, let $p_{1}=1-p_{2}\left(0\leq p_{2}\leq 1\right)$.
$$\begin{array}{ccc} E[S\left(SC_{n}\right)] & = & \left(2k-1\right)\left[-\frac{4k(k-1)}{3}n^{3}+4k\left(k-1\right)n^{2}-\frac{8k(k-1)}{3}n\right]\left(1-p_{2}\right)\\ \;\;\;\;\;\; & & +\left(2k-1\right)\left[-\frac{4k(k-2)}{3}n^{3}+4k\left(k-2\right)n^{2}-\frac{8k(k-2)}{3}n\right]p_{2}. \end{array}$$Thus,
$$\frac{\partial E(S\left(SC_{n}\right))}{\partial p_{2}}=\left(2k-1\right)\left[\frac{4k}{3}n^{3}-4kn^{2}+\frac{8k}{3}n\right]>0.$$Therefore, $E(S\left(SC_{n}\right))$ achieves the minimum value, if $p_{1}=1,p_{2}=0$, which will be $SC_{n}≅M_{n}$. This completes the proof. □

## 4. Average Values of the Indices

Denote by $\bar{\xi_{n}}$ the set of all the polygonal chains that consist of *n* regular polygons. In this section, we can characterize the average values of the Gutman index and Schultz index about $\bar{\xi_{n}}$.
$$Gut_{ave}\left(\bar{\xi_{n}}\right)=\frac{1}{|\bar{\xi_{n}}|}\sum_{G\in G_{n}}Gut\left(G\right);S_{ave}\left(\bar{\xi_{n}}\right)=\frac{1}{|\bar{\xi_{n}}|}\sum_{G\in G_{n}}S\left(G\right);$$

Let $p_{1}=p_{2}=\cdots =\frac{1}{k}$; from Theorems 1, 2, we can obtain the following result.

**Theorem** **3.**
*The average values for the indices about $\bar{\xi_{n}}$ are*

$$Gut_{avr}\left(\bar{\xi_{n}}\right)=\frac{4k^{2}\left(k+1\right)}{3}n^{3}+4k^{2}\left(k-1\right)n^{2}+\frac{4k^{2}\left(-k+2\right)}{3}n;$$


$$S_{avr}\left(\bar{\xi_{n}}\right)=\frac{2k(2k-1)(k+1)}{3}n^{3}+2k\left(2k^{2}-2k+1\right)n^{2}+\frac{2k(2k-1)(-k+2)}{3}n.$$



After verification, the equation is
$$Gut_{avr}\left(\bar{\xi_{n}}\right)=\frac{1}{k}Gut\left(\bar{M_{n}}\right)+\frac{1}{k}Gut\left(\bar{O_{n}^{1}}\right)+\frac{1}{k}Gut\left(\bar{O_{n}^{2}}\right)+\cdots +\frac{1}{k}Gut\left(\bar{P_{n}}\right);$$
$$S_{avr}\left(\bar{\xi_{n}}\right)=\frac{1}{k}S\left(\bar{M_{n}}\right)+\frac{1}{k}S\left(\bar{O_{n}^{1}}\right)+\frac{1}{k}S\left(\bar{O_{n}^{2}}\right)+\cdots +\frac{1}{k}S\left(\bar{P_{n}}\right).$$

## 5. Concluding Remarks

We mainly established the exact formulae for the expected values $E(Gut\left(SC_{n}\right))$ and $E(S\left(SC_{n}\right))$ of a random regular polygonal chain, and discussed the maximum and the minimum of $E(Gut\left(SC_{n}\right))$ and $E(S\left(SC_{n}\right))$; meanwhile, we obtained the average values $Gut_{ave}\left(\bar{\xi_{n}}\right)$ and $S_{ave}\left(\bar{\xi_{n}}\right)$, and it was finally determined that the extremal values of the indices would be obtained in this case. This paper not only proves the correctness of the previous work, but also summarizes the expected values of the Gutman index and Schultz index for all polygons containing an even number of edges. The results of this paper are helpful for the future study of topological exponentials and for building some form of mathematical model from the structure of the chemical, such as spiro compounds and alkanes whose molecular formula is $C_{n}H_{2n+2}$. Then, we may use this model to predict the activity and the physicochemical properties of other novel substances, which can provide the microscopic basis for new molecules in synthetic chemistry. It would be interesting to establish formulae for the expected values or variances of some other indices in a random irregular polygonal chain with *n* irregular polygons.

## Figures and Tables

**Figure 1 molecules-27-06838-f001:**
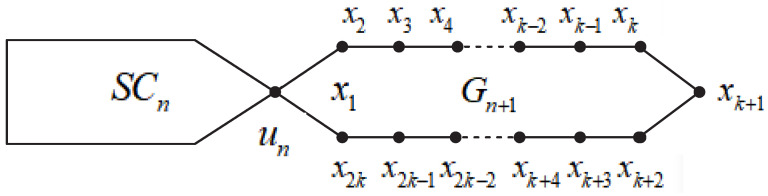
A regular polygonal chain $SC_{n+1}$ with n+1 regular polygons.

**Figure 2 molecules-27-06838-f002:**
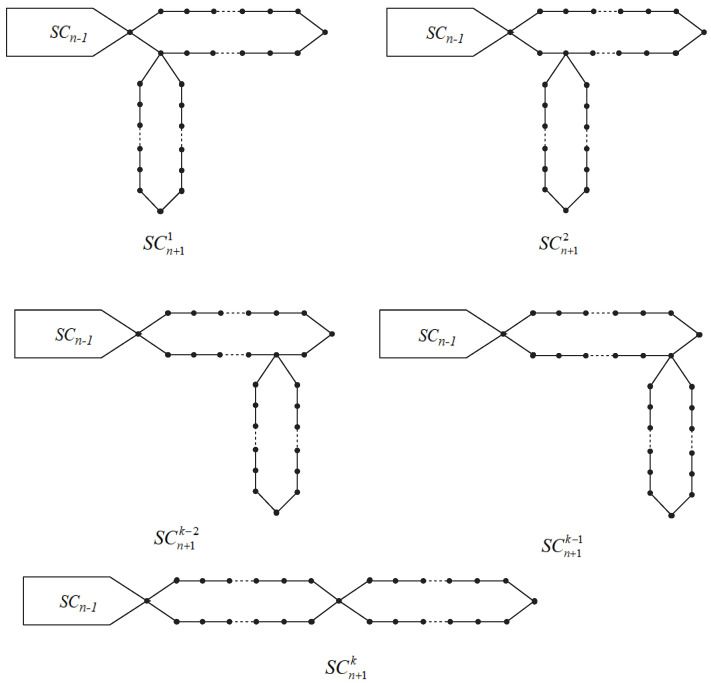
The *k* types of partial arrangements in regular polygonal chains.

## Data Availability

Not applicable.

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
