# Peer review of "The Expected Values for the Gutman Index and Schultz Index in the Random Regular Polygonal Chains"

_molecules, 2022, doi:10.3390/molecules27206838_

Round 1

Reviewer 1 Report

The paper contains interesting and novel results related to topological indices of graphs.  English needs to be improved and presentation of results are bit messy.

Author Response

Dear Reviewer:

Thank you for your decision and constructive comments on my manuscript. We have carefully considered the suggestion of Reviewer and make some changes. We have tried our best to improve and made some changes in the manuscript.

The red part that has been revised according to your comments. Revision notes, point-to-point, are given as follows:

Responds to the reviewer’s comments:

  1. Response to comment: (English needs to be improved.)

Response: Several grammatical and spelling errors have been corrected.

  1. Response to comment: (Presentation of results are bit messy.)

Response: We have put the proofs of the theorems and corollaries in their corresponding places for reading easily.

Reviewer 2 Report

In this work the authors computed the expected values of the Gutman index and Schultz index of a random regular polygonal chain with n regular polygons. Also, the average and extremal values of these indices are calculated. The paper is well written and the results are correct. The introduction contains enough background material to follow the results. The only thing i want the authors to explain if there is any relation of this topic with the spectral graph theory. If the authors can explain this, then i can recommend the paper for publication.

Author Response

Dear Reviewer:

Thank you for your letter and for the reviewers' comments concerning our manuscript entitled “The expected values for the Gutman index and Schultz index in the random regular polygonal chains”. The responds to the reviewer’s comment are as flowing:

 Response to comment: (The relation of this topic with the spectral graph theory)

Response: Topological indices are the graph invariants used in theoretical chemistry to encode molecules for the design of chemical compounds with given physicochemical properties or given pharmacological and biological activities. Many distance related topological indices (such as Wiener index, Gutman index, Kirchhoff index, Schultz index) can reflect some properties of chemical graphs. these indices are closely related to the spectra of graphs, such as Kirchhoff index can be expressed by Laplacian spectrum.

Reviewer 3 Report

Authors calculated expected values of the Gutman index and Schultz index of a random regular polygonal chain with n regular polygons.

It is interesting but relevant only when mathematical model from the structure of the chemical can be created based on values of two calculated indices.

How original is the topic? What does it add to the subject area compared with other published material?

The proposed concept is new but unfortunately no comparisons with other methods are missing. I mentioned in my review report.

Is the paper well written? Is the text clear and easy to read? Are the conclusions consistent with the evidence and arguments presented? Do they address the main question posed?

The paper is written as a theoretical paper and  no application is mentioned. Conclusions are consistent but the main question is the use of calculating indices is absent.

Q1.                  Authors mentioned that all the graphs considered here are finite, undirected and simple. Specify the meaning of simple graph.

Q2.                  Specify ordered graph.

Q3.                  Equation 1.1 needs clarification regarding its application.

Q4.                  Both indices have practical applications. Explain its implication.

Q5.                  Mention one chemical structure where indices can be used.

Author Response

Dear Reviewer:

Thank you for your letter and for the reviewers' comments concerning our manuscript entitled “The expected values for the Gutman index and Schultz index in the random regular polygonal chains”. Those comments are all valuable and very helpful for revising and improving our paper, as well as the important guiding significance to our research. We have studied comments carefully and have made correction which we hope meet with approval. Revised portions are marked in red in the paper. The main corrections in the paper and the responds to the reviewer’s comments are as flowing:

Responds to the reviewer’s comments:

  1. Response to comment: (Specify the meaning of simple graph)

Response: The definition of the simple graph has been added.

  1. Response to comment: (Specify ordered graph.)

Response: This erroneous expression has been corrected.

  1. Response to comment: (Equation 1.1 needs clarification regarding its application.)

Response: It has been added below the formula in red font.

  1. Response to comment: (Both indices have practical applications. Explain its implication.)

Response: The implication has been explained and several references have been added to explain their application.

  1. Response to comment: (Mention one chemical structure where indices can be used.)

Response: It has been added to the last paragraph in red font.

  1. Response to comment: (What does it add to the subject area compared with other published material?)

Response: This paper not only proves the correctness of the previous work, but also summarizes the expected values of Gutman index and Schultz index for all polygons containing an even number of edges. (This sentence can be found in the Concluding Remarks)